# ELMO : Efficiency via Low-precision and Peak Memory Optimization in Large Output Spaces

Jinbin Zhang[*1]   Nasib Ullah[*1]   Erik Schultheis[1 2]   Rohit Babbar[3]

## Abstract

Large output spaces, also referred to as Extreme multilabel classification (XMC), is a setting that arises, e.g., in large-scale tagging and product-to-product recommendation, and is characterized by the number of labels ranging from hundreds of thousands to millions. This means that the linear classification head, usually only a tiny fraction of the overall model, turns into the main driver for compute and memory demand. Current state-of-the-art XMC methods predominantly rely on `FP16-FP32` mixed-precision training, which we show can be unstable, and inefficient in terms of memory usage and computational overhead. Meanwhile, existing low-precision methods typically retain higher precision for the classification layer. In this work, we propose ELMO, a pure low-precision training framework for XMC models using `BFloat16` and `Float8` data types. By leveraging Kahan summation and stochastic rounding, we demonstrate that XMC models can be effectively trained entirely in `Float8`, without relying on single-precision master weights or tensor scaling. Low-precision training, combined with our proposed memory optimizations—gradient fusion and chunking—enables significant reductions in GPU memory usage. For example, we train a 3-million-label XMC model with only $6.6\,\text{GiB}$ of GPU memory, compared to the $39.7\,\text{GiB}$ required by the optimized SOTA method, Renee (Jain et al., 2023) without compromising accuracy. Code available at https://github.com/xmc-aalto/elmo.

## 1. Introduction

Large output spaces, also referred to as Extreme multilabel classification (XMC) (Bhatia et al., 2016; Babbar & Schölkopf, 2017; Prabhu et al., 2018) refers to the task of predicting a sparse subset of relevant labels from an exceedingly large set, often ranging from hundreds of thousands to millions of potential classes. XMC has gained prominence due to its applicability in real-world scenarios such as product recommendations, Wikipedia tagging, and matching search queries to advertisements. From the perspective of standard deep learning literature, this problem appears solvable using an encoder such as a CNN (Hu et al., 2014), LSTM (Hochreiter, 1997; Chung et al., 2014), or more commonly, a Transformer (Vaswani, 2017; Devlin, 2018) fine-tuned with a linear output layer in what is typically referred to as an end-to-end approach. However, contrary to other domains, where a transformer model such as BERT (Devlin, 2018) would account for a vast majority of all model parameters, in XMC, it is the classifier layer that becomes the bottleneck. For example, with an embedding dimension of 768, and three million labels, classifier weights alone would consume approximately $8\,\text{GiB}$ of memory. When accounting for gradients and optimizer states (Kingma & Ba, 2014), the memory footprint expands to around $32\,\text{GiB}$. Furthermore, loss computation in the last layer of the network, involving billions of parameters for larger datasets, also entails an enormous computational challenge.

Renee (Jain et al., 2023) shows that full end-to-end training can be feasible with appropriate memory and computational optimizations in training the model, and results in classification performance superior to the approaches employing negative sampling (Jiang et al., 2021; Zhang et al., 2021; Kharbanda et al., 2022). It exploits the fact that, particularly for neural networks in large output spaces, the loss and gradient computation using automatic differentiation engines involves maintaining large buffers for intermediate states. Therefore, as long as one can directly compute the gradients required for the backward pass, explicitly computing the loss can be forgone, hence avoiding to materialize the memory allocations for the intermediate variables altogether. In this way, Renee achieves a significant reduction in activation memory required for the classification layer.

---

[*]Equal contribution  [1]Department of Computer Science, Aalto University, Espoo, Finland [2]IST Austria [3]Department of Computer Science, University of Bath, Bath, UK. Correspondence to: Jinbin Zhang <jinbin.zhang@aalto.fi>.

*Proceedings of the 42nd International Conference on Machine Learning*, Vancouver, Canada. PMLR 267, 2025. Copyright 2025 by the author(s).

Despite the model optimizations in Renee, several memory bottlenecks remain. *Firstly*, Renee employs mixed precision (Micikevicius et al., 2018) training for the encoder, coupled with standard gradient scaling, which requires maintaining a full precision copy of the parameters. Furthermore, this approach mandates that input gradients be kept in full precision, leading to the classifier layer's gradients also being cast to full precision, which further inflates memory usage. *Secondly*, through memory snapshot analysis 3, we observe that in Renee, the order of execution in the computation graph causes memory-intensive operations to accumulate at a single point in time, leading to excessive peak memory demand. *Thirdly*, neither Renee, nor any of the label shortlisting approaches, result in a reduction in memory requirements for the classification layer weights.

To address these challenges, we first move from mixed precision to pure 16-bit training, achieving concurrent reductions in both memory and computation time. We adopt `BFloat16` (`BF16`) for gradients, leveraging its extended range to mitigate the training instability caused by potential overflows. To compensate for precision loss, we employ Kahan summation (Kahan, 1965) for the encoder optimizer and stochastic rounding (Zhang et al., 2018) for the classifier optimizer, compensating for inaccuracies and rounding bias during parameter updates. Going further, we demonstrate that classifiers can be trained in pure `Float8` (`FP8`) without scaling or mixed-precision training, as long as gradients remain in `BF16`. Finally, by integrating a `FP8` encoder from `torchao`, we achieve a nearly pure `FP8` (excluding gradients of the transformer backbone) training pipeline for XMC tasks.

Adopting pure `BF16` and `FP8` training reduces parameter memory requirements by 50–75% relative to full precision. In order to address memory accumulation in Renee, we reorganize the computation flow and decouple encoder and classifier updates, leading to more evenly distributed memory allocations throughout each training iteration and reducing peak memory usage. Furthermore, our chunking strategy for classifier updates curtails transient memory demands, and by fusing classifier gradient computation with the optimizer step in a custom Triton kernel, we effectively eliminate the need to store classifier gradients in memory. With these optimizations, the proposed method, ELMO, requires only $10.3\,\mathrm{GiB}$[1] (`BF16`) or $6.6\,\mathrm{GiB}$ (`FP8`) of memory for a 3-million-label model, significantly lower than $39.7\,\mathrm{GiB}$ necessitated by Renee.

We evaluate our low-precision training method on datasets of varying labels' sizes, demonstrating results comparable to Renee. To further assess the efficiency of our approach, we derived a new dataset with 8.6M labels from the DBLP-Citation-network V14 dataset (Tang et al., 2008; 2010; 2011;

---

[1]$1GB \approx 0.93GiB$

2012; 2007; Sinha et al., 2015). On this larger dataset, our low-precision training method demonstrates significant memory and computational savings. We believe that low-precision training will become a standard in the XMC field, given the growing demand for handling datasets with an immense number of labels.

To summarize, this paper makes the following contributions: (1) We introduce a purely low-precision training approach using `BF16` and `FP8` for models with large classification layers. (2) Combined with peak memory optimizations, this reduces memory usage by 4x–6x for a 3-million-label dataset; (3) Apart from efficiency gains, we compete with existing XMC baselines on most public datasets, illustrating that purely low-precision training can preserve similar performance; (4) We introduce LF-Paper2Keywords-8.6M, an 8.6-million-label dataset that, to our knowledge, is now the largest publicly available XMC benchmark.

## 2. Related Work

We presented discussion of Renee, the most relevant XMC method for our work, in the previous section. Next, we provide a brief overview of different categories of other XMC methods and low-precision training.

**Extreme Classification.** Initial approaches in extreme classification utilized linear classifiers with bag-of-words and TF-IDF features (Babbar & Schölkopf, 2017; 2019). As the computational cost of these methods scales linearly with the number of labels, tree-based methods were introduced to reduce computation complexity to logarithmic scale with respect to label count (Prabhu et al., 2018; Khandagale et al., 2020; Wydmuch et al., 2018; Jasinska-Kobus et al., 2021). Subsequent advancements incorporated task-specific feature learning using deep encoders atop label trees, under the assumption that joint training of both encoder and classifier layers would be computationally expensive without some form of label shortlisting for negative sampling when evaluating the loss function in the last layer (You et al., 2019; Jiang et al., 2021; Kharbanda et al., 2022; Zhang et al., 2021; Kharbanda et al., 2023). While tree-based methods provided a negative sampling mechanism, alternative approaches employed nearest-neighbor approach for negative sampling and multi-stage training to first train the encoder, followed by the classifier (Dahiya et al., 2021; 2023a; Mittal et al., 2021; Dahiya et al., 2023b; Kharbanda et al., 2025). DEXML (Gupta et al., 2024) eliminates the classifier using dual-encoder training with all-negative labels, but at the cost of increased compute and memory usage. More recently, Renee (Jain et al., 2023) demonstrated the feasibility of a fully end-to-end approach while optimizing memory consumption, revealing its potential to outperform traditional sampling-based methods. Our work advances this end-to-end approach, focusing on enhancing computational and

memory efficiency by leveraging recent developments in low-precision training.

**Low Precision Training.** With the success of foundation models (Brown et al., 2020; Achiam et al., 2023; Zhang et al., 2022; Touvron et al., 2023), there has been a strong push toward scaling both model size and training data. Nevertheless, this scalability is limited by memory and computational constraints, where low-precision training provides a notable benefit. Micikevicius et al. (2018) introduced a method for training in mixed precision (FP16-FP32) by maintaining a master copy in full precision and using loss scaling to counteract gradient underflow. The restricted range of FP16 may cause instability in large models (Rae et al., 2021; Zeng et al., 2022), necessitating a transition to BF16-FP32 mixed precision (Rae et al., 2021; Smith et al., 2022) for enhanced stability. An alternative to mixed-precision training, Zamirai et al. (2020) employs pure BF16 with methods such as Kahan summation (Kahan, 1965) and stochastic rounding (Forsythe, 1950; Croci et al., 2022) to reduce rounding errors, while COLLAGE (Yu et al., 2024) utilizes pure BF16 with a multi-component floating-point representation (Yu et al., 2022) for enhanced precision. The FP8 format is a promising next step for further reduction in precision, with early successes in using tensor scaling, as demonstrated by Micikevicius et al. (2022), and hybrid formats (Sun et al., 2019) for weights and gradients with a different exponent bias. Currently, FP8 support (torchao maintainers & contributors, 2024) is limited primarily to matmul() operations. Towards this Peng et al. (2023) introduces a comprehensive FP8 training suite, including an FP8 optimizer, communication, and distributed training capabilities. Another line of work focuses on reducing memory in Adam optimizer states (Dettmers et al., 2022; Zhao et al.) or using stateless optimizer (Lv et al., 2024). In the XMC setup, recent works have successfully achieved compression of the last layer using dynamic sparse training (Schultheis & Babbar, 2023; Ullah et al.). As an orthogonal approach, in this paper, we focus on low-precision, specifically FP8 training for XMC models, where the classifier can benefit significantly from the memory and computation efficiency of FP8 representation.

## 3. Problem setup and preliminaries

**Problem setup** For a multi-label dataset with $N$ training samples, $\mathcal{D} = \{(x_i, P_i)_{i=1}^N\}$; $L$ as the total number of labels, and $P_i \subset [L]$ denotes a subset of relevant labels associated with $x_i \in \mathcal{X}$ such that $|P_i| \ll L, \forall i$. Typically, the instances are text based, such as the contents of an article on Wikipedia or the description of a product on Amazon with labels being Wikipedia categories and frequently bought together products, respectively (Bhatia et al., 2016).

**Floating-point formats** The standard binary floating-point

representation, defined in IEEE 754 (IEE, 2019), is specified by the number of exponent bits $E$, mantissa bits $M$. the format includes special bit patterns for values like infinities, NaNs, and subnormals, which represent very small magnitudes near zero. The common 32-bit floating-point (FP32) format uses 23 mantissa bits and 8 exponent bits. Lower-precision formats, like FP16 (half-precision) and BF16 (BF16), reduce mantissa and/or exponent bits. BF16, for example, retains the 8 exponent bits of FP32 but has fewer mantissa bits, offering a similar dynamic range with reduced precision. Recently, FP8 formats have also been proposed, with 4 or 5 exponent bits and 3 or 2 mantissa bits (referred to as E4M3 and E5M2 respectively), using the IEEE 754 structure but differ in how they handle special values.

The reduction in exponent and mantissa bits in lower-precision formats introduces *quantization errors*. *Clipping error* occurs when the exponent range is insufficient, causing values outside the representable range to be "clipped" to the nearest maximum or minimum representable value. Meanwhile, *rounding error* arises from having fewer mantissa bits, which forces values to be rounded to the nearest representable value within the available precision. As a result, small weight updates are canceled due to nearest rounding (Zamirai et al., 2020).

**Stochastic Rounding and Kahan summation.** For any finite subset $F$ of the reals $\mathbb{R}$, the *stochastic rounding* (Forsythe, 1950; Croci et al., 2022; Zamirai et al., 2020) of $x \in \mathbb{R}$, is defined as follows :

$$\text{SR}(x) = \begin{cases} \lceil x \rceil & \text{with probability } p(x) \\ \lfloor x \rfloor & \text{with probability } 1 - p(x) \end{cases} \quad (1)$$

where $\lceil x \rceil = \min\{z \in F : z \geq x\}$, $\lfloor x \rfloor = \max\{z \in F : z \leq x\}$ denote rounding up and down, respectively. The probability is based on the distance $p(x) = \frac{x - \lfloor x \rfloor}{\lceil x \rceil - \lfloor x \rfloor}$. While stochastic rounding does not make any *individual* rounding operation more accurate, its result is an unbiased estimate of the true number. Thus stochastic rounding can prevent the catastrophic accumulation of rounding errors, e.g., when adding many small numbers sequentially to one large number, as is likely to happen when adding small gradient updates to the network's weights.

An alternative is to employ *Kahan summation* (Kahan, 1965). In this case, an additional buffer keeps track of the rounding error, and is used to correct subsequent additions:

$$\begin{aligned} y &\leftarrow v - c \\ c &\leftarrow ((s + y) - s) - y \\ s &\leftarrow s + y \,, \end{aligned}$$

where $s$ is the current value of the sum, $c$ is the compensation term, and $v$ is the number to be added.

At first glance, this might seem counterintuitive, as by requiring the compensation term $c$, the memory benefit of the smaller representation for $s$ is negated. However, if $s$ were kept in high precision, that would require making an additional low-precision copy for fast matrix multiplications, which is unnecessary with Kahan summation.

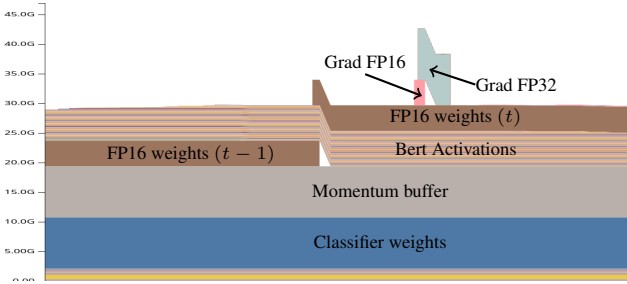

Figure 1: Memory trace of `Renee` (Jain et al., 2023) at 3 million labels & batch size 128, recorded with Pytorch profiler

**Shortcomings of mixed-precision training in Renee**
Mixed precision training (MPT) aims at improving throughput by ensuring that the most compute-intensive operations, matrix multiplications (matmul), are performed in lower-precision representations that enjoy accelerated hardware implementations. This means that weights are kept at their full precision, and a second, ephemeral low-precision copy of the weights is created for the purpose of matmul. In contrast, activations are consistently kept in lower precision, and the `Float16` gradients are cast into `Float32` during weights update.

Consequently, mixed precision training greatly benefits models with moderate parameter sizes across multiple layers (e.g., layers in the BERT-base encoder), and leads to a significant reduction in memory requirements if they are dominated by activation memory. However, its application to a single layer with extremely large parameter sizes significantly increases the peak memory consumption because it creates a second (albeit smaller) copy of the huge classification weights. Specifically, the memory trace for `Renee` presented in Figure 1 shows that the low-precision copies of the classifier weights persists for the entire step.[2] Additionally, we can see that the gradient is first calculated in 16-bit precision, but then upcast to 32 bit.[3]

As a consequence, `Renee`'s memory consumption is considerably higher than one might naïvely suspect, especially at its peak, with a total of about 40 GiB at 3 million labels. Switching to pure `BFloat16` training and fusing the SGD update so it does not need an additional buffer would im-

mediately reduce that by at least $3 \times 8\,\text{GiB}$. In the next section, we will present such a change, together with further memory optimizations, before finally reducing parameters to `FP8` for even more memory savings.

## 4. ELMO : Low-precision and Peak Memory Optimization in XMC

### 4.1. Pure 16-Bit Training

The most obvious inefficiencies visible in Figure 1 are the multiple copies of weights and gradients in different precisions. Thus, our first step towards memory-efficient training is the implementation of pure 16-bit training.

This is a bit more involved than just changing the datatype of the weights tensor, as two problems inherent in lower bit-width representations need to be overcome. `FP16` numbers, as used in `Renee`'s mixed precision training, have a much lower range of representable values than the single precision baseline. Typically, this manifests in underflows during the backward pass, leading to gradients being rounded to zero, and is mitigated by loss-scaling techniques that shift the overall range of gradients into the representable interval (Micikevicius et al., 2018).

Gradient computation for the classifier input (i.e. classifier logit gradient × classifier weights), however, involves matrix multiplications over a large inner dimension, corresponding to the number of labels. This large accumulation over labels often results in *overflow* outside the `FP16` range. At the same time, gradients of the classifier weights themselves do not grow in magnitude as the label space increases. This contrasting nature (overflow & underflow) of gradient behavior makes `Renee`'s training unstable and sensitive to both encoder and label space sizes despite using mixed precision with loss scaling (see Section 5 in Renee).

One possible solution is to adopt separate scaling factors but this requires sweeping over scale combinations (Blake et al., 2023). Even simpler, though, overflow and underflow can be easily addressed by switching from `FP16` to `BF16`, sacrificing some precision for an extended range of values that matches `FP32`. This removes the additional high-precision copy of the classifier's gradients.

The reduced precision of `BF16` comes at a price. If one just replaces the 32-bit versions of weights in the optimizer with corresponding `BF16` weights, training can no longer progress successfully. This is because round-to-nearest rounding in the optimizer update can end up canceling the update step, if it is less than half the distance to the next representable number.

There are two common ways for dealing with this problem, and we employ both. For the classifier weights, where memory efficiency is paramount, we use stochastic round-

---

[2]This particular problem could easily be fixed by `del`'ing them after the calculating of the classifier layer's gradient.

[3]Renee official code

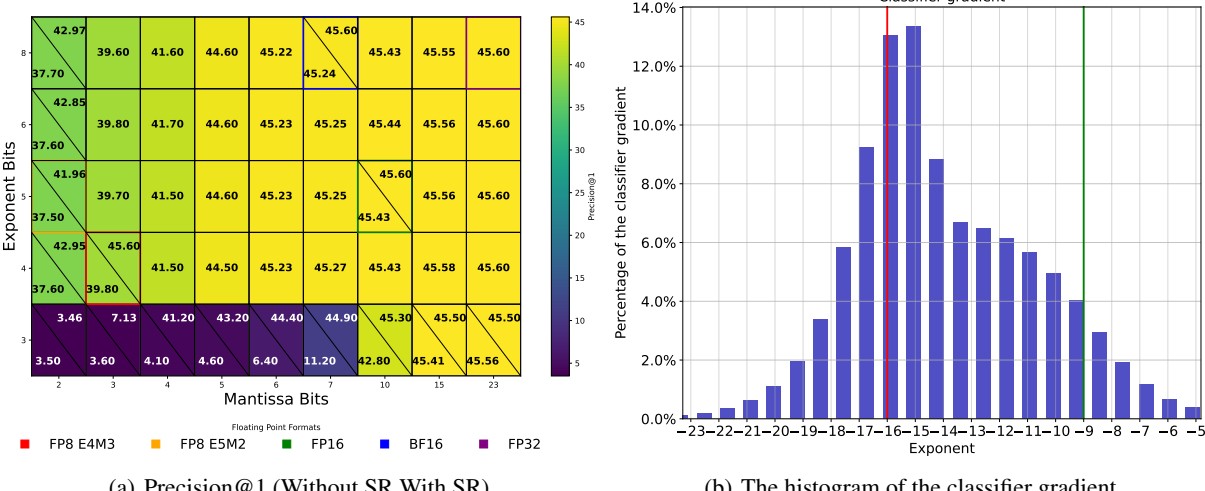

(a) Precision@1 (Without SR With SR)    (b) The histogram of the classifier gradient

Figure 2: Figure 2(a): Precision@1 performance at different exponent and mantissa bit patterns for the classifier weights. The numbers above diagonal is the performance when stochastic rounding is applied. Figure 2(b): histogram of classifier gradients. Around 20% of gradients become zero in `Float8 E5M2`([-16, 15]), while nearly 90% of gradients drop to zero in `Float8 E4M3`([-9, 8]). The gradients are sampled from the training of LF-AmazonTitles-131K.

ing (Forsythe, 1950; Croci et al., 2022; Zamirai et al., 2020) as defined in (1). This reduces the amount of memory required for classifier weights to one third of the `Renee` setting and halves the optimizer state.The weights of the `Bert` encoder, in contrast, only make up a tiny fraction of the total memory consumption. Therefore, we opt to use *Kahan summation* (Kahan, 1965) for the encoder's parameters.

## 4.2. Architectural improvements

After fixing the most egregious inefficiencies stemming from standard mixed-precision training, we now turn to further architectural enhancements to reduce the (peak) memory consumption of the training algorithm.

A noticeable chunk of the remaining memory is taken up by the momentum buffer for the classifier weights. Our experiments showed that momentum is not required so we remove it entirely, switching to pure large-learning-rate SGD for the classifier layer.

Finally, there remains the spike in memory consumption due to the classification layer's gradient. There are several ways this could be addressed. A first option would be to change the order of operations and calculate this gradient only at the end of the backward pass, when activation memory for the `Bert` encoder has already been freed. While this does not actually reduce the memory requirements for this operation, it is moved to a point in time with less memory pressure, thus reducing peak memory consumption.

Reorganizing computation flow helps alleviate peak memory

consumption; however, as label size increases, memory allocated for classifier logits and gradients becomes the primary bottleneck. We employ chunking (Rabe & Staats, 2021; Hsu et al., 2024) for classifier parameters: We first execute the encoder's forward pass, then divide the labels into $k$ equal-sized chunks, processing the classifier's forward pass, backward pass, and optimization step sequentially for each chunk. The encoder's backward pass and update occur after all classifier operations are complete. This reduces transient memory requirements for classifier operations by a factor of $k$. We use between 3 and 8 chunks for XMC datasets, observing no impact on training latency (see Appendix F).

### 4.3. 8-Bit Training

For further memory savings, we turn towards reducing the number of bits allocated to each parameter further. To that end, we first investigate how much precision and dynamic range are required for the training process, by simulating floating-point numbers with a specific number of mantissa and exponent bits.

Figure 2(a), which illustrates the performance for LF-AmazonTitles-131K dataset in terms of precision@$k$, shows that 3 exponent bits provide sufficient range to represent classifier weights, whereas 2 bits are not enough. In the mantissa, we see significant degradation starting to set in as it shrinks below 6 bits, but this can be counteracted by stochastic rounding, which recovers the original performance. This leads us to choose the `E4M3`, which is directly supported in `Hopper`, `Ada`, and `Blackwell` GPUs, for representing

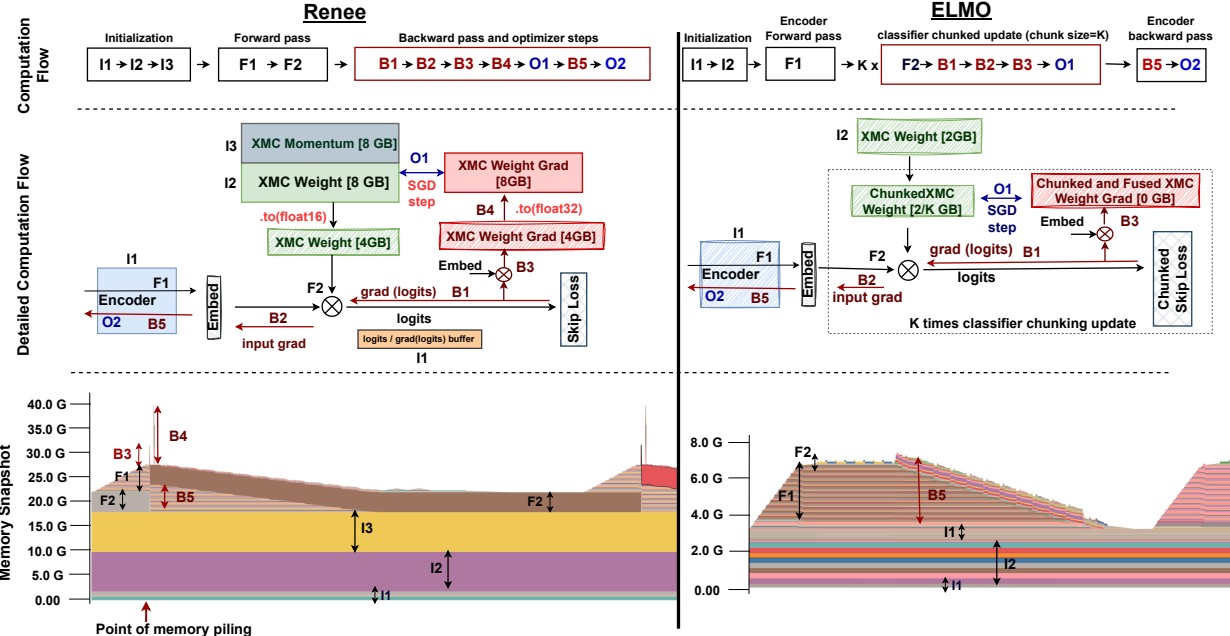

Figure 3: GPU memory comparison between Renee and the proposed approach (ELMO) at various instances during one round of forward and backward pass. Note the difference in the scale of Y-axis in the two cases. The graphic was created using Pytorch memory viz utility.

classifier weights; notably, we do not introduce any additional tensor scaling (Micikevicius et al., 2022), the native dynamic range of E4M3 is sufficient, c.f. also Figures 5(a) and 5(b).

In contrast, Figure 2(b) indicates that exponents of around 20% of the gradients exceed even the representable range of FP8 E5M2, necessitating the use of BF16. To address the different precision requirements for weights (in FP8) and gradients (BF16), we cast classifier inputs from BF16 to FP8 E4M3 when computing logits, as FP8 sufficiently covers the inputs range (Figure 5(b)). We then perform a matrix multiplication between FP8 E4M3 inputs and FP8 E4M3 weights, but obtain logits in BF16 for higher-precision gradient computation. For input gradient computation, which involves matrix multiplication between FP8 weights and BF16 logits, we developed a Triton (Tillet et al., 2019) kernel to manage this, avoiding additional HBM memory usage due to data type differences.

To reduce the memory overhead caused by the gradient, ELMO fuses gradient computation and SGD updates into a single Triton (Tillet et al., 2019) kernel, optimized for execution entirely in SRAM. Within this kernel, classifier weights, logits, and inputs are loaded from GPU memory into SRAM. Classifier weight gradients are computed via a matmul operation between the logits and inputs. The classifier weights are then updated directly in SRAM using SGD with stochastic rounding before being written back to

GPU memory. This eliminates the need to store classifier gradients in GPU memory, reducing its memory footprint to nearly zero.

Further, we take on the recently developed torchao (torchao maintainers & contributors, 2024) framework for the encoder training, which leverages FP8 training for transformers, reducing the encoder's activation memory requirements compared to BF16 training. By combining FP8 training for both the encoder and classifier, end-to-end training of FP8 XMC models becomes feasible.

### 4.4. Comparison of ELMO and Renee

Finally, let us consider training a 3 million label model with a batch size of 128, using Bert-base as an encoder with 768 embedding dimensions. How much memory do the different parts of the models take? We use the above Figure 3, which shows a comparison of the memory snapshot for the two approaches, and corresponding order of operations during initialization, forward and backward pass.

At initialization (denoted by I1, I2 etc. in Figure 3), Renee allocates memory for encoder parameters, its optimizer states, and a buffer to store logit gradients for the labels in last layer. For this example setting, parameters and momentum for the classifier layer amount to $8\,\mathrm{GiB}$ [4] each, the logit

---

[4] $8 \approx \frac{768 \times 2,812,281 \times 4}{1024 \times 1024 \times 1024} = \frac{\text{embed\_dim} \times \text{num\_labels} \times \text{num\_bytes\_per\_fp32}}{1024 \times 1024 \times 1024}$

Table 1: Statistics of XMC Datasets with and without Label Features. This table presents a comparison across various datasets, detailing the total number of training instances ($N$), unique labels ($L$), number of test instances ($N'$), average label count per instance ($\overline{L}$), and average data points per label ($\hat{L}$).

| Dataset | $N$ | $L$ | $N'$ | $\overline{L}$ | $\hat{L}$ |
|---|---|---|---|---|---|
| **Datasets without Label Features** | | | | | |
| Wiki-500K | 1,779,881 | 501,070 | 769,421 | 4.75 | 16.86 |
| AmazonTitles-670K | 485,176 | 670,091 | 150,875 | 5.39 | 5.11 |
| Amazon-670K | 490,449 | 670,091 | 153,025 | 5.45 | 3.99 |
| Amazon-3M | 1,717,899 | 2,812,281 | 742,507 | 36.17 | 31.64 |
| **Datasets with Label Features** | | | | | |
| LF-AmazonTitles-131K | 294,805 | 131,073 | 134,835 | 5.15 | 2.29 |
| LF-WikiSeeAlso-320K | 693,082 | 312,330 | 177,515 | 4.67 | 2.11 |
| LF-AmazonTitles-1.3M | 2,248,619 | 1,305,265 | 970,237 | 22.2 | 38.24 |
| LF-Paper2Keywords-8.6M | 2,020,621 | 8,623,847 | 2,020,621 | 9.03 | 2.12 |

buffer consumes $687\,\mathrm{MiB}$. ELMO gets rid of the momentum buffer altogether, and allocates weights in either 16-bit ($4\,\mathrm{GiB}$) or 8-bit ($2\,\mathrm{GiB}$). Due to chunking, the size of the logits' gradients gets divided by the number of chunks, in this case 8, but stays in 16-bit representation for both `BF16` and `FP8` training, leading to $86\,\mathrm{MiB}$. Additionally, the `Bert` model and its optimizer states are allocated. These are the same size for both ELMO and Renee, amounting to $\approx 1.2\,\mathrm{GiB}$. In total, ELMO allocates $3.2\,\mathrm{GiB}$ ($5.2\,\mathrm{GiB}$ for `BF16`) at initialization, 70-80% reduced compared to Renee's $17.9\,\mathrm{GiB}$.

During the forward propagation steps (denoted as F1, F2 etc. in Figure 3), activation memory accumulates for the `Bert` transformer, $4.6\,\mathrm{GiB}$ in `BF16` and $3\,\mathrm{GiB}$ in `FP8` mixed precision. For the classifier layer, Renee also needs to create the `FP16` copy of its weights, an additional $4\,\mathrm{GiB}$. During the backward pass of Renee (B1, B2 etc.), gradients are allocated ($4\,\mathrm{GiB}$) and converted to `FP32` $8\,\mathrm{GiB}$, with all these allocations stacking up to a peak memory consumption of $39.7\,\mathrm{GiB}$. Much more efficiently, ELMO does not need to make a copy of the classifier weights, nor does it materialize classifier gradients. However, the FP8 encoder uses additional buffers of $0.5\,\mathrm{GiB}$, bringing the peak memory consumption to $6.6\,\mathrm{GiB}$.

## 5. Contributed Dataset

**Motivation.** Performance evaluation for XMC algorithms traditionally relies on public benchmarks (Bhatia et al., 2016), where the largest dataset contains 3 million labels. Consequently, many recent large-scale XMC experiments have employed proprietary datasets (Mehta et al., 2024; Jain et al., 2023; Dahiya et al., 2021). A more expansive public dataset would facilitate the exploration and comparison of modern, efficient training strategies, while also revealing

bottlenecks that become significant at larger label sizes.

**LF-Paper2Keywords-8.6M.** We curated LF-Paper2Keywords-8.6M using the DBLP-Citation-network V14 dataset (Tang et al., 2008; 2010; 2011; 2012; 2007; Sinha et al., 2015). This dataset comprises the titles and abstracts of research papers sourced from DBLP, ACM, and MAG (Microsoft Academic Graph), with each paper's keywords serving as labels. The dataset includes 8.6 million labels and can support tasks such as automated keyword suggestion and paper recommendation for research articles. Complete dataset details are provided in Table 1.

## 6. Experiments and Discussion

**Datasets.** We validated our approach on a broad suite of XMC datasets spanning both non–label-feature-based (Amazon-670K, Wiki-500K, Amazon-3M, AmazonTitles-670K) and label-feature-based (LF-AmazonTitles-131K, LF-WikiSeeAlso-320K, LF-AmazonTitles-1.3M, and our newly curated LF-Paper2Keywords-8.6M). All except LF-Paper2Keywords-8.6M are publicly accessible through the Extreme Classification Repository (Bhatia et al., 2016). Detailed descriptions are provided in the Table 1.

**Baselines and Evaluation Metrics.** We compared our method with two categories of baselines: *(i) Sampling-Based XMC*, focusing on Transformer-based methods (e.g., LightXML (Jiang et al., 2021), CascadeXML (Kharbanda et al., 2022)) for non-label-feature datasets and NGAME (Dahiya et al., 2023a) and DEXML (Gupta et al., 2024) for label-feature datasets; *(ii) End-to-End XMC*, with Renee (Jain et al., 2023) as the principal baseline. Following standard XMC practices, we evaluate all methods using top-$k$ metrics, specifically Precision@$k$ and its propensity-scored variant. A detailed overview is given in the Appendix A.

Table 2: Comparison of the precision performance of our proposed ELMO method with state-of-the-art XMC methods on the Wiki-500K, AmazonTitles-670K, Amazon-670K, and Amazon-3M datasets, as well as the label feature datasets LF-WikiSeeAlso-320K and LF-AmazonTitles-1.3M. Bold indicates the best results, and underline indicates the second-best. $M_{\mathrm{tr}}$ denotes peak training memory and METHOD-N denotes a method with $N$ ensembles.

| Method | Encoder | P@1 | P@3 | P@5 | $M_{\mathrm{tr}}$ (GiB) | Epoch Time (mm:ss) | P@1 | P@3 | P@5 | $M_{\mathrm{tr}}$ (GiB) | Epoch Time (mm:ss) |
|---|---|---|---|---|---|---|---|---|---|---|---|
| | | | | | Wiki-500K | | | | | AmazonTitles-670K | |
| LIGHTXML | BERT-Base | 76.19 | 57.22 | 44.12 | 15.72 | 147:27 | 41.7 | 37.3 | 34.2 | 13.99 | 19:02 |
| CASCADEXML | BERT-Base | 77.0 | 58.3 | 45.1 | 18.8 | 50:00 | 42.1 | 37.5 | 34.1 | 22.3 | 11:32 |
| LIGHTXML-3 | BERT-Base | 77.78 | 58.85 | 45.57 | 15.72 | 3×147:27 | 43.1 | 38.7 | 35.5 | 13.99 | 3×19:02 |
| CASCADEXML-3 | BERT-Base | 78.39 | 59.86 | 46.49 | 18.8 | 3×50:00 | 43.5 | 39.0 | 35.6 | 22.3 | 3×11:32 |
| RENEE | BERT-Base | **78.69** | 60.03 | 46.46 | 12.69 | 18:37 | 43.78 | 39.17 | 35.91 | 12.46 | 2:04 |
| ELMO (BF16) | BERT-Base | 78.61 | **60.04** | **46.59** | 7.21 | 17:01 | 44.3 | 39.7 | **36.4** | 5.12 | 1:47 |
| ELMO (FP8) | BERT-Base | 78.39 | 59.64 | 46.12 | **5.01** | **11:28** | **44.39** | **39.75** | 36.43 | **3.37** | **1:18** |
| | | | | | Amazon-670K | | | | | Amazon-3M | |
| LIGHTXML | BERT-Base | 47.3 | 42.2 | 38.5 | 11.5 | 53:30 | - | - | - | OOM | - |
| CASCADEXML | BERT-Base | 48.5 | 43.7 | 40.0 | 18.3 | 16:46 | 51.3 | 49.0 | 46.9 | 87.0 | 90:00 |
| LIGHTXML-3 | BERT-Base | 49.1 | 44.17 | 40.25 | 11.5 | 3×53:30 | - | - | - | OOM | - |
| CASCADEXML-3 | BERT-Base | 50.22 | 45.20 | **41.45** | 18.3 | 3×16:46 | 53.10 | 50.64 | 48.49 | 87.0 | 3×90:00 |
| RENEE | BERT-Base | 50.6 | 45.16 | 41.13 | 11.91 | 7:14 | 52.6 | 49.7 | 47.43 | 39.7 | 29:58 |
| ELMO (BF16) | BERT-Base | **50.7** | **45.27** | 41.29 | 5.29 | 6:00 | **53.4** | **50.9** | **48.8** | 10.39 | 25:15 |
| ELMO (FP8) | BERT-Base | 50.34 | 44.91 | 40.97 | **3.3** | **4:05** | 52.73 | 50.38 | 48.27 | **6.6** | **18:02** |
| | | | | | LF-WikiSeeAlso-320K | | | | | LF-AmazonTitles-1.3M | |
| LIGHTXML | Distil-BERT | 34.5 | 22.31 | 16.83 | 24.46 | 61:09 | - | - | - | OOM | - |
| NGAME | Distil-BERT | 45.72 | 29.61 | 22.06 | 16.63 | 57:19 | 54.99 | 48.09 | 43.11 | 11.03 | 40:09 |
| DEXML | Distil-BERT | 46.78 | 30.32 | 22.59 | 38.6 | 242:51 | **58.4** | - | **45.46** | 75.53 | 1054:21 |
| RENEE | Distil-BERT | 47.86 | 31.91 | 24.05 | 13.89 | 10:50 | 56.04 | **49.91** | 45.32 | 19.9 | 9:12 |
| ELMO (BF16) | Distil-BERT | 47.84 | **31.99** | **24.12** | 6.57 | 10:08 | 56.14 | 49.86 | 45.25 | 6.61 | 9:10 |
| ELMO (FP8) | Distil-BERT | **47.88** | 31.92 | 24.09 | **5.2** | **6:22** | 54.97 | 48.41 | 43.82 | **4.31** | 17:44 |

Table 3: Precision performance comparison on LF-Paper2Keywords-8.6M dataset. Other XMC baselines do not scale to 8.6 million label size.

| Method | P@1 | P@3 | P@5 | $M_{\mathrm{tr}}$(GiB) |
|---|---|---|---|---|
| FLOAT32 | 43.60 | 32.13 | 26.02 | 58.44 |
| RENEE | 17.65 | 11.78 | 9.23 | 105.64 |
| ELMO(BF16) | **45.4** | **33.58** | **27.18** | 18.8 |
| ELMO(FP8) | 43.4 | 31.59 | 25.38 | **9.02** |

**Implementation Details.** Low-precision training with chunking is implemented using the PyTorch framework (Paszke et al., 2017). For the encoder, we used AdamW (Loshchilov & Hutter, 2017) optimizer with Kahan summation provided by the optimi library[5]. The in-place SGD optimizer with stochastic rounding for the classifier is im-

plemented via custom Triton and CUDA kernels. We also employ Triton kernels for the FP8 classifier and gradient fusion. All BF16 experiments are conducted on an A100 GPU, while FP8 experiments are run on an H100 (Table 2, 3 and 8) and RTX 4060Ti (Table 5) GPU. Further details on implementation and hyperparameters are provided in the Appendix G.

**Empirical Performance.** Table 2 reports Precision@$k$ results for ELMO on non–label-feature datasets, benchmarked against state-of-the-art XMC baselines. Notably, ELMO significantly reduces peak memory usage—its FP8 variant achieves a 6 times reduction relative to Renee (Jain et al., 2023) (the current end-to-end optimized approach) and a 13 times reduction compared to sampling-based methods. Despite these efficiency gains, ELMO shows competitive or improved Precision@$k$ performance. For label-feature datasets, we apply the standard augmentation (Kharbanda et al., 2024; Jain et al., 2023) strategy summarized in Table 2 (LF-WikiSeeAlso-320K, LF-AmazonTitles-1.3M) and Table 8 (LF-AmazonTitles-131K). Our results demonstrate

---

[5]https://github.com/warner-benjamin/optimi

that ELMO maintains robust performance in both FP8 and BF16 modes, underscoring the versatility of our approach.

Finally, on our newly introduced LF-Paper2Keywords-8.6M dataset as shown in Table 3, ELMO provides substantial memory savings, requiring only 18.8 GiB (`BF16`) or 9.02 GiB (`FP8`), compared to 105 GiB for Renee. Notably, `BF16` ELMO even outperforms the `Float32` baseline, likely benefiting from the regularization effects of stochastic rounding (Ozkara et al., 2025). `FP8` also delivers performance close to that of `Float32`. Renee underperforms, likely due to gradient overflow in the classifier input caused by its use of `FP16` data types.

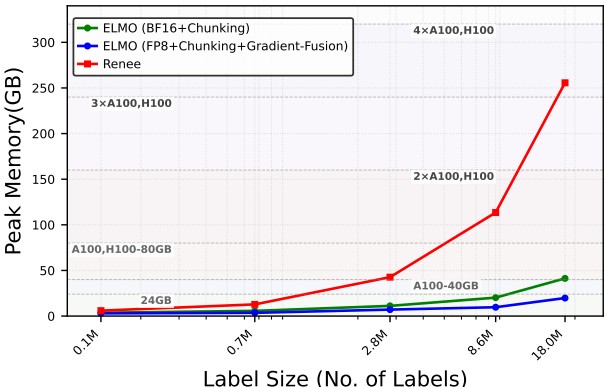

Figure 4: Comparison of peak GPU memory usage across varying label sizes for ELMO and Renee (Jain et al., 2023).

**Label Size vs Peak GPU Memory.** Figure 4 plots peak GPU memory usage as label sizes grow from 131K (LF-AmazonTitles-131K) to 18 million. While the largest public dataset has 3 million labels, we introduce one with 8.6 million. Beyond 8.6 million, random labels are appended to measure peak usage. `BF16` and `FP8` significantly reduce GPU memory compared to Renee (Jain et al., 2023); for instance, at 3 million labels, ELMO (`FP8`) lowers memory by 6 times, increasing to 11 times at 8.6 million and 13 times at 18 million.

**`BFloat16` vs. `Float8` Encoders.** For our encoder, we can also use `BF16` instead of the `FP8` encoder from `torchao` (torchao maintainers & contributors, 2024) , while the XMC classifier continues to use `FP8`. We compare the performance with different precision settings in Table 4, which shows similar precision but longer epoch time for FP8 due to the overhead in the FP8-BF16 mixed precision recipe.

## 7. Conclusion

We present a low-precision training framework with `BFloat16` and `Float8` for XMC models, moving be-

Table 4: Comparison of precision performance between the `BFloat16` and `Float8` encoders with the classifier fixed at `Float8`.

| Encoder | P@1 | P@3 | P@5 | $M_{\mathrm{tr}}$ (GB) | Epoch Time (mm:ss) |
|---|---|---|---|---|---|
| | LF-AmazonTitles-1.3M | | | | |
| BF16 | **55.08** | **48.47** | **43.87** | 5.50 | **17:26** |
| FP8 | 54.97 | 48.41 | 43.82 | **4.63** | 17:44 |
| | Amazon-3M | | | | |
| BF16 | 52.60 | 50.23 | 48.11 | 8.51 | **15:56** |
| FP8 | **52.73** | **50.38** | **48.27** | **7.16** | 18:02 |

yond the conventional practice of relaxing the classification layer's precision. In contrast to `FP16-FP32` MPT which can be memory-inefficient and unstable, our method is robust and, with gradient fusion and chunking, reduces memory by 4x–6x for a 3-million-label dataset. Notably, these efficiency gains do not compromise performance; our approach competes with existing XMC baselines on most public datasets. Furthermore, we introduce a new XMC dataset LF-Paper2Keywords-8.6M with 8.6 million labels, which, upon its release, will be the largest publicly available XMC dataset. We anticipate that this new dataset will spur further innovation in extreme classification. While our proposed training recipe does not require (micro-)tensor scaling, our investigation in 2(a) indicates that future work aiming to further reduce representation bitwidth to FP6 or FP4 datatypes will have to take such strategies into account.

## Impact Statement

We do not anticipate any negative societal impact of our work. It is expected that the affordable training methodologies developed in this work will further enable the exploration of similar methodologies for other deep networks which are more affordable and easily accessible to a broader research community.

## Acknowledgements

We acknowledge the support of the Academy of Finland (Research Council of Finland) via grants 347707 and 348215 and the support of computational resources provided by the Aalto Science-IT project, and CSC IT Center for Science, Finland.

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

## A. Baselines and Evaluation Metrics

We compare our method with deep XMC methods with mainly transformer encoder.

- **LightXML** (Jiang et al., 2021): The method employs a transformer encoder to concurrently train both the retriever and ranker, which incorporates dynamic negative sampling to enhance the model's efficacy.

- **CascadeXML** (Kharbanda et al., 2022): This method separates the feature learning of distinct tasks across various layers of the Probabilistic Label Tree (PLT) and aligns them with corresponding layers of the transformer encoder.

- **NGAME** (Dahiya et al., 2023a): NGAME enhances transformer-based training for extreme classification by introducing a negative mining-aware mini-batching technique, which supports larger batch sizes and accelerates convergence by optimizing the handling of negative samples.

- **Renee** (Jain et al., 2023): The Renee model employs an integrated end-to-end training approach for extreme classification, using a novel loss shortcut for memory optimization and a hybrid data-model parallel architecture to enhance training efficiency and scalability.

- **DEXML** (Gupta et al., 2024): The DEXML model aims to eliminate the need for an explicit classifier, instead relying solely on dual-encoder-based training with all negative labels within each batch. While the motivation to remove the classifier (often the primary bottleneck) is sound, their approach ultimately incurs higher computational and memory costs compared to methods that retain a classifier.

To evaluate the performance of our Extreme Multi-label Classification model, which incorporates low-precision training, we use a set of metrics designed to provide a comprehensive analysis of both overall and label-specific model performance. The primary metrics we employ is Precision at $k$ (P@$k$), which assess the accuracy of the top-$k$ predictions. Additionally, we incorporate Propensity-Scored Precision at $k$ (PSP@$k$) to gauge the uniformity of the model's effectiveness across the diverse range of labels typical in XMC problems.

**Precision at $k$ (P@$k$):**  Precision at k is the fundamental metric for evaluating the top-$k$ predictions in XMC applications such as e-commerce product recommendation and document tagging:

$$P@k(y, \hat{y}) = \frac{1}{k} \sum_{\ell \in \text{top}_k(\hat{y})} y_\ell \qquad (2)$$

where $y$ is the true label vector, $\hat{y}$ is the predicted score vector, and $\text{top}_k(\hat{y})$ identifies the indices with the top-$k$ highest predicted scores.

**Propensity-Scored Precision at $k$ (PSP@$k$):**  Given the long-tailed label distribution in many XMC datasets, PSP@k incorporates a propensity score $y_l$ to weight the precision contribution of each label, thereby emphasizing the tail labels' performance:

$$PSP@k(y, \hat{y}) = \frac{1}{k} \sum_{\ell \in \text{top}_k(\hat{y})} \frac{y_\ell}{p_\ell} \qquad (3)$$

where $p_l$ corresponds to the propensity score for the label $y_l$ (Jain et al., 2016).

---

**Algorithm 1** `Float8` XMC classifier

```
class FP8Classifier(nn.Module):
    def XMC_update(self, X, labels):
        # X [b, m] inputs for the classifier
        # Rows and cols of the positive labels
        X_gradient = torch.zeros_like(X)
        rows, cols = labels[:,0], labels[:,1]
        for i in range(self.num_chunks):
            # self.W[i] [n, m], the classifier weights of
                one chunk
            logits = matmul_fp8(
              self.W[i], X.t().to(torch.float8_e4m3fn)
            ) # The logits are in BF16
            rows_i, cols_i = filter_chunk_i_labels(rows,
                cols)
            logits = logits.sigmoid_()
            logits[cols_i, rows_i] -= 1
            X_gradient += large_k_matmul(logits,
                                         self.W[i])

            fuse_update(self.W[i], self.lr, logits, X, X.
                shape[0], self.bs, self.bs, self.bs)
        return X_gradient

    def fuse_update(W, lr, logits, X, K, bk, bm, bn):
        # the pseudo code of the triton kernel
        grad = tl.zeros((bm, bn), dtype=FP32)
        for _ in range(0, K, bk):
            x = load_block_from_HBM(X).to(BF16)
            l = load_block_from_HBM(logits).to(BF16)
            grad += block_matmul(l, x)
        w = load_block_from_HBM(W).to(FP32)
        w = w - lr*grad
        w = stochastic_rounding_to_FP8(w)
        write_to_HBM(w, W)
```

---

Table 5: Epoch level training statistics on the RTX 4060 Ti. $M_{\text{tr}}$ denotes peak training memory.

| Dataset | Epoch Time (mm:ss) | $M_{\text{tr}}(GB)$ |
|---|---|---|
| LF-AmazonTitles-1.3M | 57:36 | 5.45 |
| Amazon-3M | 121:17 | 8.46 |
| LF-Paper2Keywords-8.6M | 229:24 | 10.49 |

## B. Background of Skipping Loss Computation

For the XMC problems in this paper, we use the binary cross-entropy loss. Following Renee (Jain et al., 2023), we

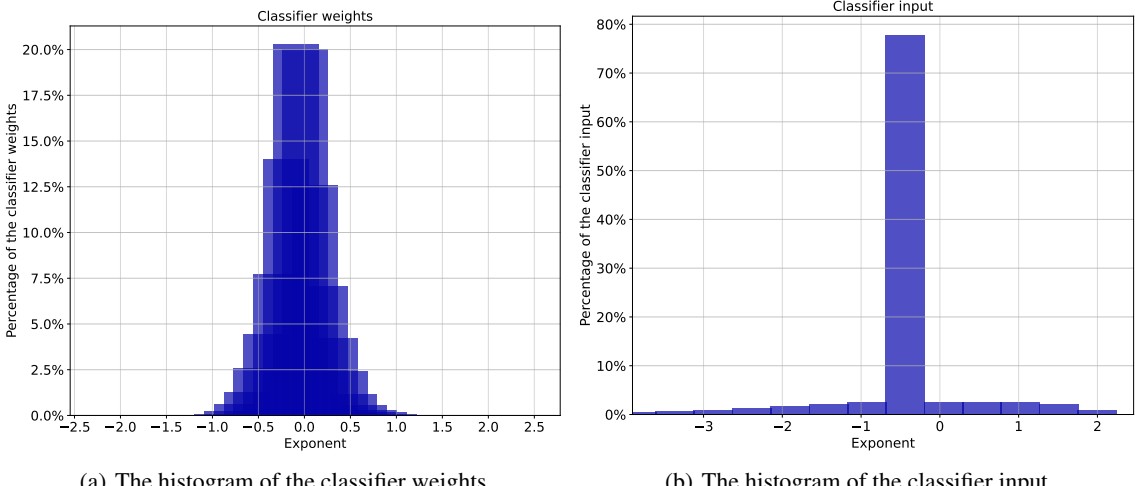

(a) The histogram of the classifier weights      (b) The histogram of the classifier input

Figure 5: Figures 5(a) and 5(b) show that most weights and classifier inputs fall within the exponent range of `FP8` E4M3 ([-9, 8]), even without quantization. The weights and inputs are sampled from the training of LF-AmazonTitles-131K.

Table 6: Precision performance with Post-Hoc classifier refinement (on top of `Float8` checkpoint from ELMO ) and Kahan summation for head labels (20% head) on the LF-AmazonTitles-1.3M. $M_{\mathrm{tr}}$ denotes peak training memory.

|  | P@1 | p@3 | P@5 | $M_{\mathrm{tr}}(GiB)$ |
|---|---|---|---|---|
| Renee | 56.04 | 49.91 | 45.32 | 19.9 |
| BF16 (ELMO) | 56.14 | 49.86 | 45.25 | 6.61 |
| Float8 (ELMO) | 54.97 | 48.41 | 43.82 | 4.31 |
| Post-Hoc | 55.4 | 48.87 | 44.34 | 4.31 |
| Head Kahan | 55.6 | 49.38 | 44.88 | 4.65 |

apply skipping loss computation, so the gradient for the classifier input is given by:

$$\mathrm{InputGrad} = \left(\frac{1}{1+e^{-y}} - Y\right) * W$$

where $y$ represents the logits, $Y$ is the ground truth, and $W$ denotes the classifier weights.

Similarly, the gradient for $W$ is:

$$\nabla W = \left(\frac{1}{1+e^{-y}} - Y\right) * X$$

where $X$ is the input embedding for the classifier.

We refer to $\left(\frac{1}{1+e^{-y}} - Y\right)$ as the "classifier logit gradient" in our paper.

## C. Deployment on Commodity Hardware

Although our main `Float8` experiments use H100 GPUs, we also demonstrate that the approach runs efficiently on commodity hardware, such as the GeForce RTX 4060 Ti. As shown in Table 5, the 4060 Ti uses slightly more memory, primarily due to the `torch.ao` package not yet fully supporting `Float8` on commodity hardware. As a result, we use a `BF16` encoder while still applying `Float8` to the XMC classifiers.

## D. Precision Recovery for Sensitive Applications

For applications where recovering the last bit of accuracy is critical, two potential practical mitigation strategies that still operate within similar memory budgets are:

1. **Post-hoc Classifier Refinement:** A simple approach is to fine-tune the classifier in higher precision on top of an ELMO-trained (low-precision) model using frozen encoder features. This allows a partial recovery of the lost precision while staying within a constrained memory budget by loading only subsets of labels at a time. This strategy introduces an additional training phase and hyperparameters to be tuned for the second stage.

2. **Kahan summation for head labels:** To address accuracy drops without additional training stages, we also outline another approach that leverages label statistics inherent in XMC tasks. By exploiting the long-tailed label distribution, one can apply Kahan summation with `BF16` compensation only to the top-p% most fre-

Table 7: Comparison of Propensity based Precision@k for different XMC methods. The best results are denoted by bold and second best results are denoted by underline.

| Method | PSP@1 | PSP@3 | PSP@5 | PSP@1 | PSP@3 | PSP@5 | PSP@1 | PSP@3 | PSP@5 |
|---|---|---|---|---|---|---|---|---|---|
| | Wiki-500K | | | AmazonTitles-670K | | | Amazon-670K | | |
| ATTENTIONXML | 30.69 | 38.92 | 44 | 24.24 | 26.43 | 28.39 | 30.29 | 33.85 | 37.13 |
| XR-TRANSFORMER | 32.1 | 39.41 | 43.75 | - | - | - | 29.21 | 33.49 | 37.65 |
| CASCADEXML | 31.25 | 39.35 | 43.29 | - | - | - | 30.23 | 34.93 | 38.79 |
| RENEE | 32.9 | 42.31 | 46.78 | 27 | 31.1 | 34.89 | **31.45** | **36.16** | **40.15** |
| ELMO (BF16) | **33.32** | **42.56** | **47.03** | **28.62** | **32.13** | **35.27** | 30.84 | 35.69 | 40.06 |
| ELMO (FP8) | 32.40 | 41.68 | 46.17 | 28.24 | 31.88 | 35.26 | 30.57 | 35.33 | 39.67 |
| | Amazon-3M | | | LF-WikiSeeAlso-320K | | | LF-AmazonTitles-1.3M | | |
| LIGHTXML | - | - | - | 17.85 | 21.26 | 24.16 | - | - | - |
| XR-TRANSFORMER | - | - | - | 25.18 | 30.13 | 33.79 | 20.06 | 24.85 | 27.79 |
| NGAME-2 | - | - | - | **33.83** | **37.79** | 41.03 | 29.18 | 33.01 | 35.36 |
| RENEE | 14.39 | 17.47 | 19.80 | 32.02 | 37.07 | 40.9 | 28.54 | 33.38 | 36.14 |
| ELMO (BF16) | 15.65 | 19.05 | 21.6 | 31.65 | 37.08 | **41.04** | **30.38** | **34.59** | **37.09** |
| ELMO (FP8) | **16.06** | **19.48** | **21.98** | 31.87 | 36.98 | 40.90 | 26.72 | 31.58 | 34.46 |

Table 8: Precision and propensity scored precision comparison of ELMO with state of the art XMC baselines on LF-AmazonTitles-131K dataset. The best results are denoted by bold and second best are by underline. $M_{\mathrm{tr}}$ denotes peak training memory.

| Method | P@1 | P@3 | P@5 | PSP@1 | PSP@3 | PSP@5 | $M_{\mathrm{tr}}$(GiB) | Epoch Time (mm:ss) |
|---|---|---|---|---|---|---|---|---|
| LIGHTXML | 35.6 | 24.15 | 17.45 | 25.67 | 31.66 | 36.44 | 16.79 | 10:27 |
| NGAME | 44.69 | 29.89 | 21.21 | 38.81 | 44.4 | 49.43 | 11.03 | 5:15 |
| DEXML | 42.52 | - | 20.64 | - | - | 48.7 | 29.22 | 14:08 |
| RENEE | **46.05** | **30.81** | **22.04** | **39.08** | **45.12** | **50.48** | 5.53 | 0:33 |
| ELMO (BF16) | 45.6 | 30.6 | 21.9 | 38.84 | 45.02 | 50.39 | 3.41 | 0:31 |
| ELMO (FP8) | 45.45 | 30.53 | 21.87 | 38.75 | 44.98 | 50.41 | **2.75** | **0:22** |

quent labels. This approach selectively boosts precision@k, with minimal memory overhead, approximately 2×p% (where p% is memory for p% label parameters in `Float8` ) more than the `Float8` baseline. Importantly, this strategy preserves end-to-end training and avoids the complexity of multi-stage pipelines. For example, as shown in Table 6, on AmazonTitles-1.3M with top 20% head labels, this method achieves a competitive performance as Reene with a total classifier memory footprint of just 4.99 GB, still significantly below the `BF16` baseline (6.61 GB).

## E. Tail Label Performance

Table 7 compares the propensity scored precision@k values, which is an indication of tail label performance. It would be interesting to explore the effect of tail label performance

when going down in bit-width. Similar to the precision@k performance, our approach shows competitive performance with existing state-of-the-art XMC baselines, showcasing low precision training can be robust to tail labels. Performance of LF-AmazonTitles-131k is shown in Table 8.

## F. Chunking Classifier Update: Latency vs Peak Memory

Table 10 shows the latency (epoch time) vs peak GPU memory usage comparison for different chunk sizes. The training run is with ELMO BF16 data types for 3 million data size with batch size 128 on H100 GPU. We see chunking doesn't affect the latency until some point. In fact, we see the latency improves as the chunking increases from 1 to 8.

Table 9: The hyper-parameters of the `BFloat16` and `Float8` models. Dropout is the embedding dropout and Encoder LR and XMC LR are the learning rates for the encoder and classifiers . WD denotes weight decay.

| Dataset | Encoder | Batch Size | Seq. Length | Dropout | Encoder LR | XMC LR | Epochs | Warmup | WD (Encoder, XMC) |
|---|---|---|---|---|---|---|---|---|---|
| **Dataset without Label Features** | | | | | | | | | |
| Wiki-500K (`BF16`) | BERT-Base | 128 | 128 | 0.65 | 0.00002 | 0.05 | 35 | 1000 | (0.01, 0.0001) |
| Wiki-500K (`FP8`) | BERT-Base | 128 | 128 | 0.65 | 0.00002 | 0.15 | 70 | 1000 | (0.01, 0.0001) |
| AmazonTitles-670K (`BF16`) | BERT-Base | 256 | 32 | 0.7 | 0.00005 | 0.05 | 100 | 1000 | (0.01, 0.0001) |
| AmazonTitles-670K (`FP8`) | BERT-Base | 256 | 32 | 0.75 | 0.00005 | 0.05 | 150 | 1000 | (0.01, 0.0001) |
| Amazon-670K (`BF16`) | BERT-Base | 64 | 128 | 0.75 | 0.00002 | 0.06 | 100 | 500 | (0.01, 0.0001) |
| Amazon-670K (`FP8`) | BERT-Base | 64 | 128 | 0.7 | 0.00002 | 0.05 | 150 | 1000 | (0.01, 0.0001) |
| Amazon-3M (`BF16`) | BERT-Base | 128 | 128 | 0.6 | 0.00005 | 0.05 | 90 | 10000 | (0.001, 0.001) |
| Amazon-3M (`FP8`) | BERT-Base | 128 | 128 | 0.65 | 0.00002 | 0.05 | 150 | 10000 | (0.001, 0.001) |
| **Dataset with Label Features** | | | | | | | | | |
| LF-AmazonTitles-131K (`BF16`) | Distil-BERT | 512 | 32 | 0.84 | 0.00001 | 0.1 | 100 | 15000 | (0, 0) |
| LF-AmazonTitles-131k (`FP8`) | Distil-BERT | 512 | 32 | 0.85 | 0.00001 | 0.05 | 100 | 5000 | (0.01, 0.0001) |
| LF-WikiSeeAlso-320K (`BF16`) | Distil-RoBERTa | 128 | 256 | 0.75 | 0.00002 | 0.08 | 100 | 5000 | (0.01, 0.0001) |
| LF-WikiSeeAlso-320K (`FP8`) | Distil-RoBERTa | 128 | 256 | 0.75 | 0.00002 | 0.08 | 100 | 5000 | (0.01, 0.0001) |
| LF-AmazonTitles-1.3M (`BF16`) | Distil-BERT | 512 | 32 | 0.65 | 0.000005 | 0.05 | 100 | 5000 | (0.0001, 0.0001) |
| LF-AmazonTitles-1.3M (`FP8`) | Distil-BERT | 512 | 32 | 0.6 | 0.000005 | 0.2 | 150 | 15000 | (0.01, 0.0001) |
| LF-Paper2Keywords-8.6M (`BF16`) | Distil-BERT | 128 | 128 | 0.70 | 0.00002 | 0.05 | 12 | 5000 | (0.01, 0.0001) |
| LF-Paper2Keywords-8.6M (`FP8`) | Distil-BERT | 128 | 128 | 0.70 | 0.00002 | 0.05 | 12 | 5000 | (0.01, 0.0001) |

Table 10: Peak memory vs Latency (in terms of epoch time) for different chunk size when chunking classifier update is used with `BF16` on Amazon-3M dataset.

| Chunk Size | Epoch Time (mm:ss) | Peak Mem (GiB) |
|---|---|---|
| 1 | 13:22 | 14.74 |
| 2 | 12:20 | 14.40 |
| 4 | 12:12 | 12.22 |
| 8 | **11:09** | 11.13 |
| 16 | 11:23 | 10.59 |
| 32 | 12:39 | 10.32 |
| 64 | 14:19 | **10.20** |
| 128 | 19:44 | **10.20** |

weights before performing matrix multiplication. This operation is handled within the Triton kernel, ensuring there is no additional memory overhead. We apply this for LF-AmazonTitles-1.3M.

## G. Hyper-parameters and Implementation Details

We detail the hyper-parameters of the `BF16` and `FP8` models in the table 9.

## H. Low-Memory Dropout for Classfier

To improve the classifier's robustness, we apply dropout (Wan et al., 2013) to it. However, a limitation of traditional dropout is its requirement for a copy of the classifier's weights, doubling memory consumption. To address this, we implement dropout directly within the matrix multiplication process. After loading the classifier's weights from HBM into SRAM, we apply dropout to the loaded

