# OpenReview forum: "ELMO : Efficiency via Low-precision and Peak Memory Optimization in Large Output Spaces"
_ICML.cc/2025/Conference — ICML 2025 poster_

### Official Review · Reviewer_VYeF · 2025-03-10

**Overall Recommendation:** 3

**Summary:**

This paper proposes a collection of quantization techniques to reduce memory usage of extreme classification (where the number of classes is large). Proposed techniques include stochastic rounding, Kahan summation, FP8 weight, FP16 gradient, chunking, etc, leading to several folds of memory usage reduction compared to previous methods.

## Update after rebuttal

The authors rebuttal have addressed my concerns. I raised my score.

**Claims And Evidence:**

yes

**Essential References Not Discussed:**

N/A

**Experimental Designs Or Analyses:**

yes

**Methods And Evaluation Criteria:**

yes

**Other Comments Or Suggestions:**

I think the term "8-bit training" and "16-bit training" should be usage with caution. By default I suppose 16-bit training is PyTorch amp, and 8-bit training is TransformerEngine, which conducts matrix multiplications in low-precision.

**Other Strengths And Weaknesses:**

Strength:
- The paper has a significant memory reduction compared to previous techniques.
- The memory profiling is thorough.

Weaknesses:
- Novelty: the paper is largely applying a collection of existing techniques to a specific problem. Key techniques such as stochastic rounding and chunking are quite standard. This paper looks more like an engineering optimization rather than research.
- While the paper proposes some specific techniques for extreme classification, there might be some more out-of-the-box general methods which can also achieve the goal. For example, 8-bit optimizer and per-block quantization [1].

[1] Dettmers T, Lewis M, Shleifer S, et al. 8-bit Optimizers via Block-wise Quantization[C]//International Conference on Learning Representations.

**Questions For Authors:**

NA

**Relation To Broader Scientific Literature:**

See strengths and weaknesses.

**Theoretical Claims:**

N/A

---

> ### Author Rebuttal · Authors · 2025-03-31
>
> > Novelty: the paper is largely applying a collection of existing techniques to a specific problem. Key techniques such as stochastic rounding and chunking are quite standard. This paper looks more like an engineering optimization rather than research.
>
> Please check our response to reviewer M72L on how our approach and setup differs from the similar innovations which are proposed largely in the context of LLMs.
>
>
>
> > While the paper proposes some specific techniques for extreme classification, there might be some more out-of-the-box general methods which can also achieve the goal. For example, 8-bit optimizer and per-block quantization [1].
>
> Most prior work on low-memory optimizers (e.g., Adam-8Bit, GaLore) focuses on reducing the memory footprint of optimizer states. However, when using pure SGD—as we do for the classifier—these methods offer no memory savings and can even increase memory usage due to unnecessary state tracking (see response to Reviewer  M72L on “How does ELMO compare to other FP8 training frameworks?”).
> Beyond memory, these stateful optimizers (e.g., Adam) also underperform in terms of accuracy when applied to the classifier. This is due to the extreme sparsity of tail-label updates in XMC, which makes momentum-based states noisy and less effective. In contrast, stateless SGD is both more memory-efficient and yields better performance in this setting.
> The table below shows an ablation study comparing optimizers for the classifier on the LF-AmazonTitles-131K and AmazonTitles-670K datasets. The findings in Table 5 of Renee paper also corroborate this.
>
> LF-AmazonTitles-131K
> | Classifier optimizer | P@1 | P@3 | P@5 |
> | --- | --- | --- | --- |
> | SGD+SR (ELMO)| 45.5 | 30.5 | 21.9|
> | Adam+SR | 45.2 |  30.4| 21.8 |
> |  Adam-8bit| 42.9 |28.7  | 20.5 |
>
> AmazonTitles-670K
> | Classifier optimizer | P@1 | P@3 | P@5 |
> | --- | --- | --- | --- |
> | SGD+SR (ELMO)| 44.4 | 39.7 | 36.4 |
> | Adam+SR | 43.6 |  39.2| 36.0 |
> |  Adam-8bit| 42.8 |38.6  | 35.3 |

---

> > ### Comment · Reviewer_VYeF · 2025-04-07
> >
> > The reviewer would like to thank the authors for the rebuttal. However, I still have concerns on the novelty. As per Reviewer M72L, there are highly related previous work such as LOMO. While the proposed method has smaller memory footprint than LOMO, the claimed advantages such as not needing to store a whole tensor, and the hybrid approach of SGD and Adam, both seems can be implemented quite straightforwardly from LOMO. Additionally, sharding approach such as Zero might also solve the memory problem? I still do not quite see the reason that the problem cannot be solved with off-the-shelf approaches.

---

> > > ### Author Response · Authors · 2025-04-07
> > >
> > > Thank you very much for your response. We respectfully disagree that our approach is highly related to LOMO and can be implemented straightforwardly from it. We have clearly distinguished our contributions from LOMO, and would like to reiterate a few key points:
> > > - LOMO fuses the gradient update step along with the optimizer step and here fusion means it is fused for each layer but still materialises the gradients in GPU memory and then does the optimizer step for that layer.  Please check this for better understanding about fusion in  LOMO (https://pytorch.org/tutorials/intermediate/optimizer_step_in_backward_tutorial.html). In our case materializing (temporary storage in GPU memory) the gradient would consume enormous memories and a significant bottleneck (as we explained in the main paper and previous response). We don't materialize the gradient at all for FP8 classifiers (uses SRAM to keep the gradients and update the optimizer step while staying in SRAM) and this fusion is generally called kernel level fusion [1]. Also we would like to mention that the specific kernel level fusion strategies (e.g for Attention Layer[1], Loss calculation[2] and in our case specific for XMC classifier) is generally not considered as a straightforward extension.
> > >
> > > [1] Dao, Tri, et al. "Flashattention: Fast and memory-efficient exact attention with io-awareness." Advances in neural information processing systems 35 (2022): 16344-16359.
> > >
> > > [2] Wijmans, Erik, et al. "Cut your losses in large-vocabulary language models." arXiv preprint arXiv:2411.09009 (2024).
> > >
> > >
> > > - Precision Differences (FP16 vs. FP8): LOMO uses FP16 precision, whereas our method employs FP8. Transitioning to FP8 significantly impacts training dynamics, stability, and convergence behavior, differentiating our method fundamentally from LOMO.
> > >
> > >
> > > - Limitations of Mixed-Precision in LOMO: LOMO uses the FP32-FP16 mixed precision for parameters (https://github.com/OpenLMLab/LOMO/blob/main/lomo/src/lomo.py#L100) and gradients (https://github.com/OpenLMLab/LOMO/blob/main/lomo/src/lomo.py#L86) , inherently facing the same memory bottleneck highlighted in Renee which we discussed in Figure 1 and Shortcomings of mixed-precision training in Renee in section 3. On the 8.6M-label dataset, **LOMO** requires approximately **64 GB** of memory for the XMC classifier, whereas our **FP8-based ELMO** approach significantly reduces this to only **6.3 GB**.
> > >
> > >
> > > - Hybrid Optimizer Strategy: LOMO exclusively uses SGD, which underperforms for the encoder component in XMC scenarios. Given the encoder's minimal memory footprint compared to the classifier, we adopt a hybrid optimizer strategy (SGD for the classifier and AdamW for the encoder). While this is indeed a notable difference, we do not claim this optimizer strategy itself as our  novelty.
> > >
> > > Additionally, we have provided detailed explanations on why off-the-shelf methods designed to optimize optimizer-state memory are not directly applicable in our scenario. We have also thoroughly outlined distinctions between our FP8 methodology and other existing FP8 techniques.
> > > Regarding Zero, it primarily addresses distributed multi-GPU setups, whereas our focus here is explicitly on single-GPU training. Hence, Zero is not applicable in our context.

---

### Official Review · Reviewer_uk61 · 2025-03-12

**Overall Recommendation:** 4

**Summary:**

This paper considers the problem of extreme classification where given an input, it is to be categorized into a few categories among a large set of possible categories. Full one-vs-all classifier training is an expensive approach to solve this problem, but has been shown to give best results and be possible to scale to 100M label space by Renee paper. This paper identifies some of the challenges in the Renee implementation, primarily, mixed precision training leading to a big memory footprint. It suggests fp8 based training and relevant optimizations around it to mitigate rounding errors. This yields a much more memory compact one-vs-all classifier training implementation that scales to tens of millions of labels in a reasonable amount of compute.

**Claims And Evidence:**

Yes.

**Essential References Not Discussed:**

No

**Experimental Designs Or Analyses:**

Yes. The experiments are done in the standard evaluation setting of extreme classification as well as compared against SoTA baseliens which make the evaluation protocol trustworthy.

**Methods And Evaluation Criteria:**

Yes.

**Other Comments Or Suggestions:**

- It would help to visualize Figure 4 in a systematic ablation form i.e. have multiple plots with optimizations being added sequentially (perhaps in sorted order of their impact)
- Table captions should be more verbose, as a reader it's easier if the details to parse a table are given in the caption

**Other Strengths And Weaknesses:**

### Strengths
- Paper is well presented, easy to read
- The motivation is clear and straightforward
- A new dataset is contributed which will help in evaluating larger scale XMC research

### Weakness
- Scope is limited to extreme classifier based approach which is not very broadly applicable

**Questions For Authors:**

1. Do these training optimizations easily transfer to dual encoder training, specifically to something like the one-vs-all dual-encoder training in DEXML?
2. How do baseline (CascadeXML, DEXML, etc) methods perform on the new LF-Paper2Keywords-8.6M dataset?

**Relation To Broader Scientific Literature:**

Extreme classification is a practical problem in recommendation scenarios. Although there has been a lot of work on generic deep learning training optimizations, but because of the special nature of an extreme classification solution (i.e. a wide final classifier) makes it open for specialized training optimizations, which is the core contribution of this paper.

**Theoretical Claims:**

N/A

---

> ### Author Rebuttal · Authors · 2025-03-31
>
> > Scope is limited to extreme classifier based approach which is not very broadly applicable.
>
> Beyond the immediate applications in tagging, search and recommendation systems, large output spaces are becoming largely prevalent in modern LLMs [1,2,3] with increasing vocabulary sizes such as 256K tokens for Gemma2-2B. This presents another scenario where our Float8 training paradigm could be effectively applied.
>
> [1] Wijmans, et al. “Cut Your Losses in Large-Vocabulary Language Models", arXiv:2411.09009
>
> [2] Tao, Chaofan, et al. "Scaling laws with vocabulary: Larger models deserve larger vocabularies." arXiv preprint arXiv:2407.13623 (2024).
>
> [3] Yu, Da, et al. "Scaling Embedding Layers in Language Models." arXiv preprint arXiv:2502.01637 (2025).
>
> > It would help to visualize Figure 4 in a systematic ablation form i.e. have multiple plots with optimizations being added sequentially (perhaps in sorted order of their impact)
>
> Thanks for the suggestion! Section 4.4, along with Figure 3, already provides a detailed analysis of each optimization's contribution to memory usage, so adding an additional ablation study in the main paper might be somewhat repetitive, especially given the strict 8-page limit. However, we would include the systematic ablation visualization you've described as supplementary material in the appendix.
>
> > Table captions should be more verbose, as a reader it's easier if the details to parse a table are given in the caption
>
> Thanks for pointing this out! We'll make the table captions more detailed to improve readability.
>
> > Do these training optimizations easily transfer to dual encoder training, specifically to something like the one-vs-all dual-encoder training in DEXML?
>
> Our method focuses on optimizing the classifier in XMC, while DEXML’s main memory bottleneck comes from computing label embeddings via the encoder. One can use an FP8 encoder (e.g., via torch.ao) and store label embeddings in FP8 to enable FP8 matmul for scoring. However, gradient fusion offers limited benefit in this setting, since label embeddings are produced by the encoder and not updated like classifier weights.
>
> > How do baseline (CascadeXML, DEXML, etc) methods perform on the new LF-Paper2Keywords-8.6M dataset?
>
> On the LF-Paper2Keywords-8.6M dataset, we found that baseline methods like DEXML and CascadeXML did not scale to our multi-GPU training setups, including both 4×A100-80GB and 4×H100-80GB nodes. For example, in CascadeXML, the classifier parameters alone require approximately 25 GB of memory. When accounting for optimizer states (gradients and two momentum buffers), the total memory footprint becomes roughly 100 GB (25 + 25 + 2×25). Under Distributed Data Parallel (DDP), these parameters and states are replicated across GPUs, offering no effective memory savings.  Thank you for pointing this out and we will add these remarks in the appendix of the updated version of the paper.

---

### Official Review · Reviewer_M72L · 2025-03-15

**Overall Recommendation:** 2

**Summary:**

This paper introduces ELMO, a low-precision training framework designed to optimize memory and computation for Extreme Multi-label Classification (XMC), where the classification layer dominates memory and compute costs. Key results include: 1) 6× memory reduction compared to Renee (previous SOTA) for 3M-label models, 2) comparable accuracy compared to FP16/FP32 training on XMC tasks, 3) peak memory reduced to 6.6GiB (FP8), compared to 39.7GiB in Renee.

**Claims And Evidence:**

1. ELMO significantly reduces memory usage in XMC models. Supported by empirical experiments, which shows that ELMO reduces peak memory by 75% compared to Renee.

2. FP8 with stochastic rounding and Kahan summation stabilizes training for ELMO. Also, ELMO achieves training efficiency gains without sacrificing performance, as shown in Table 1.

**Essential References Not Discussed:**

Fused update has already been proposed in early literature, such as below

1. https://arxiv.org/abs/2306.09782
2. https://arxiv.org/abs/2403.03507

**Experimental Designs Or Analyses:**

Experiments in the paper covers:

1. Memory efficiency compared to Renne across datasets
2. Ablation on chunking and fused updates
3. Trade off between FP8 and FP16

**Methods And Evaluation Criteria:**

ELMO introduces several techniques:

1. FP8 Weights and BF16 Gradients
2. Stochastic Rounding and Kahan Summation
3. Chunking and Gradient Fusion
4. Reordered Computation Flow

**Other Comments Or Suggestions:**

None

**Other Strengths And Weaknesses:**

Lack of novelty: most of techniques (FP8, fused update, etcs) have already been proposed and widely used in recent works. Applying them to a new learning task (XMC) might not have sufficient novelty.

**Questions For Authors:**

1. How does ELMO compare to other FP8 training frameworks? such as FP8-LM
2. How much computational overhead does gradient fusion and chunking introduce?

**Relation To Broader Scientific Literature:**

related to low-precision training, memory-efficient training

**Theoretical Claims:**

There is no theoretical claims included in the paoer

---

> ### Author Rebuttal · Authors · 2025-03-31
>
> > Fused update has already been proposed in early literature
>
> Thank you for pointing out these references. While related in motivation, our approach differs in key aspects:
> - LOMO: LOMO performs layer-wise fused updates by materializing gradients, applying optimizer steps, and releasing memory (https://pytorch.org/tutorials/intermediate/optimizer_step_in_backward_tutorial.html) before backward pass through next layer. While this reduces memory relative to full gradient accumulation, it still requires storing the entire gradient of the largest layer at each step. For example, in XMC with 3M labels, the classifier gradient alone requires ~8 GB, compared to an accumulated memory footprint of ~8.7 GB ( 0.7 GB related to non classifier). In contrast, our method applies fused updates using a kernel that avoids materializing the classifier gradient. This reduces peak memory usage to just ~0.7 GB, offering a significantly more efficient solution. We also differ in the choice of optimizers: while both methods use SGD, we adopt a hybrid approach—SGD for the classifier, and AdamW for the encoder achieving better performance-memory trade-offs.
>
> - GaLore: GaLore reduces memory by projecting gradients into a low-rank space, thereby compressing optimizer states. However, this is not directly applicable in our setting as the classifier —responsible for the majority of the memory footprint—is trained with plain SGD, without momentum or adaptive states. Moreover, SGD performs slightly better than AdamW for classifier training (as shown below in the table of the response to reviewer 4), an observation which corroborates the findings in Table 5 of Renee paper.
>
> We appreciate the reference provided and will include it in the final version.
>
> > Lack of novelty: most of techniques (FP8, fused update, etcs) have already been proposed and widely used in recent works. ...
>
> While most of the mentioned techniques are tailored for optimizing large language models (LLMs), these do not directly apply to the setting of large output spaces. For a more detailed explanation based on concrete instances on how our approach differs, please refer to our response to the next question.
>
> > How does ELMO compare to other FP8 training frameworks? such as FP8-LM
>
> The existing FP8 frameworks primarily retain higher precision for the classification layer. For instance, FP8-LM uses float16 data types. In contrast, in extreme classification, reducing classifier parameter size is crucial. We demonstrate that using FP8 data types for classifiers is viable, achieving competitive performance without the need for tensor scaling, thus eliminating additional overhead or expensive hyperparameter sweeps (in case of loss scaling).
>
> Existing FP8 training frameworks for LLM—including FP8-LM, Transformer Engine, COAT, and torch.ao—focus on reducing memory by quantizing optimizer states or employ mixed-precision strategies that reduce activation memory. However, in XMC, performance is best when using stateless optimizers like SGD. As a result, these methods are not directly applicable, and the dominant memory bottleneck becomes the classifier weights and their gradients, not the optimizer states. For example, on the LF-Paper2Keywords-8.6M dataset, the classifier in FP8-LM would consume approximately 37 GB of memory, whereas our approach requires only 6 GB. Additionally, mixed-precision training—commonly used in LLMs—is particularly harmful in XMC due to increased memory usage and should be avoided in practice.
>
> In summary, our contributions differ from existing FP8 frameworks in several important ways:
> (1) We apply low precision directly to the classifier, unlike typical LLM-focused methods;
> (2) We show that FP8 classifiers can achieve strong performance in XMC without relying on scaling techniques;
> (3) We demonstrate that mixed-precision strategies are counterproductive in XMC due to memory overhead and are better avoided.
>
> > How much computational overhead does gradient fusion and chunking introduce?
>
> We have included an ablation study of chunk size versus latency in Table 7. We observe that increasing the chunk size initially reduces epoch time up to an optimal point, after which epoch time begins to increase. We selected a chunk size below this optimal threshold.
>
> Gradient fusion introduces no computational overhead and further improves efficiency by reducing gradient-related I/O operations. Without gradient fusion, the number of I/O operations for the classifier update with sgd is: num\_labels×dim×6 (loading classifier weights for logits and input gradient computation, saving/loading classifier gradient, updating classifier) + num\_labels×batch\_size×7 (saving logits into HBM, loading it for input and classifier gradient computation, loading for sigmoid and its gradient computation). With gradient fusion, this reduces to: num\_labels×dim×4 + num\_labels×batch\_size×7.

---

### Official Review · Reviewer_QZwr · 2025-03-16

**Overall Recommendation:** 3

**Summary:**

This paper presents ELMO, an efficient training method designed for solving extreme multilabel classification (XMC) problems by leveraging low-precision computation. The key techniques that ELMO leverages are pure 16-bit training, classifier parameter chunking, and 8-bit training to reduce memory usage and improve training speed. Comprehensive experimental results have been demonstrated to justify the effectiveness of the proposed ELMO method, as well as showing that it outperforms the prior state-of-the-art Renee framework by a significant margin.

**Claims And Evidence:**

The major claims of speed-up and memory savings made by ELMO are well supported by the experimental evaluations.

**Essential References Not Discussed:**

N/A

**Experimental Designs Or Analyses:**

The experimental designs and analyses make sense.

**Methods And Evaluation Criteria:**

Both the proposed methods and the evaluation criteria make sense for solving XMC with compute and memory efficiency.

**Other Comments Or Suggestions:**

Please see details provided in "Other Strengths And Weaknesses".

**Other Strengths And Weaknesses:**

Strengths:
- This paper is well-written and well-motivated in general.
- Exploring computation and memory-efficient methods for solving the XMC problem is a promising research direction.
- The experimental results of this paper seem to be convincing.

Weaknesses:
- It seems that using the ELMO method inevitably causes a precision drop compared to FP32 methods, which can potentially hurt the method's practicality for precision-sensitive applications.
- It is also not clear how to mitigate the precision/accuracy drop caused by the ELMO method.
- FP8 computation is only supported for GPUs using the Hopper architecture and newer, which limits the usability of the method for users with older generations of GPUs.

**Questions For Authors:**

- Is there a way to write a fused CUDA kernel for the proposed ELMO method to further improve computation and memory efficiency?

**Relation To Broader Scientific Literature:**

The key contributions of this paper are related to low-precision efficient training more broadly.

**Theoretical Claims:**

There are no theoretical claims in this paper.

---

> ### Author Rebuttal · Authors · 2025-03-31
>
> > It seems that using the ELMO method inevitably causes a precision drop compared to FP32 methods, which can potentially hurt the method's practicality for precision-sensitive applications.
>
> On 5 out of 7 datasets in the paper, pure FP8/FP16 training with ELMO already achieves similar prediction performance as full/mixed precision methods such as Renee. The precision drop observed in the LF-Paper2Keywords-8.6M dataset was due to a preprocessing error on our part. Specifically, as a result of this oversight, we omitted the paper titles from the input to the ELMO model, whereas the baseline Float32 Renee utilized this information. After correcting this by concatenating the title and abstract as input to ELMO, the updated results for FP8 ELMO (in table below) are now very close to the Float32 Renee model.
>
>
> | LF-Paper2Keywords-8.6M | P@1 | P@3 | P@5 |
> | --- | --- | --- | --- |
> | Float32 | 43.6 | 32.13  | 26.02 |
> |  Previous FP8|  39.93| 29.88 |24.33  |
> | Updated FP8 | 43.4 |  31.59| 25.38 |
>
> On the LF-AmazonTitles-1.3M dataset, the difference is within ~1.5% point in P@k, which can further be reduced to be within ~0.5% as mentioned in the answer to the next question.
>
> > For precision-sensitive applications, it is also not clear how to mitigate the precision/accuracy drop caused by the ELMO method.
>
> For such applications where recovering the last bit of accuracy is critical, two potential practical mitigation strategies that still operate within similar memory budgets are:
>
> (i) Post-hoc Classifier Refinement: A simple approach is to fine-tune the classifier in higher precision on top of an ELMO-trained (low-precision) model using frozen encoder features. This allows a partial recovery of the lost precision while staying within a constrained memory budget by loading only subsets of labels at a time. This strategy introduces an additional training phase and hyperparameters to be tuned for the second stage.
>
> (ii) Kahan summation for head labels: To address accuracy drops without additional training stages, we also outline another approach that leverages label statistics inherent in XMC tasks. By exploiting the long-tailed label distribution, one can apply Kahan summation with BF16 compensation only to the top-P% most frequent labels. This approach selectively boosts precision@k, with minimal memory overhead—approximately 2×P% (where P% is memory for P%  label parameters in FP8) more than the FP8 baseline. Importantly, this strategy preserves end-to-end training and avoids the complexity of multi-stage pipelines. For example, on AmazonTitles-1.3M with top 20% head labels, this method achieves a competitive performance as Reene with a total classifier memory footprint of just 4.99 GB, still significantly below the BF16 baseline (6.61 GB).
>
> The results of the above mentioned mitigation strategies are shown in the table below.
>
>
> | LF-AmazonTitles-1.3M | P@1 | P@3 | P@5 | Memory(GB) |
> | --- | --- | --- | --- | --- |
> | Renee | 56.04 | 49.91 |45.32  | 19.9 |
> | Original FP8 | 54.97 | 48.41 | 43.82 | 4.63 |
> |  Post Hoc| 55.4 | 48.87 | 44.34 | 4.63 |
> | Head Kahan |55.6  | 49.38 | 44.88 | 4.99 |
>
> > FP8 computation is only supported for GPUs using the Hopper architecture and newer, which limits the usability of the method for users with older generations of GPUs.
>
> FP8 computation is supported on Ada, Hopper, and Blackwell tensor cores. Notably, our model enables Float8 XMC classifiers to run on commodity GPUs, including the RTX 4000 series. For example, we successfully ran ELMO on an RTX 4060 with a memory footprint of just 10.49GB (on LF-Paper2Keywords-8.6M), achieving a training time of 3.5 hours per epoch. In contrast, Renee (58.4GB) and other state-of-the-art models cannot run on these affordable consumer GPUs.
>
>
> For users with GPUs that do not natively support FP8 computation, we can store classifier weights in FP8 and upcasts them chunkwise to BF16 during forward and backward passes. This reduces memory usage while maintaining compatibility with older hardware. For example, on the LF-Paper2Keywords-8.6M dataset, ELMO with this strategy runs successfully on an A100 GPU with a memory footprint of 13.5 GB, compared to 20 GB for a pure BF16 and 9.6 GB for the FP8 version.
>
> > Is there a way to write a fused CUDA kernel for the proposed ELMO method to further improve computation and memory efficiency?
>
> A fused CUDA kernel would maintain similar memory efficiency to our fused Triton kernel. However, additional low-level warp- and thread-based optimizations might further improve computational performance.

---

### Decision · Program_Chairs · 2025-05-01

**Decision:**

Accept (poster)

**Comment:**

The ELMO paper proposes a low-precision training framework for XMC (extreme multi-label classification) models, using BFLOAT16 and FLOAT8 data types. It demonstrates that XMC models can be effectively trained entirely in FLOAT8, without relying on single-precision master weights or tensor scaling, by leveraging Kahan summation and stochastic rounding. The method also incorporates memory optimizations like gradient fusion and chunking.

The paper is well-written and well-motivated. ELMO significantly reduces memory usage in XMC models, with empirical evidence showing a 75% reduction compared to Renee. FP8 training with stochastic rounding and Kahan summation stabilizes training for ELMO, achieving training efficiency gains without sacrificing performance.

On the negative side, the paper lacks novelty, as most of the techniques (FP8, fused update, etc.) have been proposed before. The paper is however solid and open-source release (especially of the Triton kernel) would significantly strengthen the contribution.

Overall, we recommend weak acceptance.